# Plasma Fluorinated Nano-SiO_2_ Enhances the Surface Insulation Performance of Glass Fiber Reinforced Polymer

**DOI:** 10.3390/nano13050906

**Published:** 2023-02-28

**Authors:** Qijun Duan, Guowei Xia, Yanze Song, Guohua Yin, Yuyao Zhong, Jun Xie, Qing Xie

**Affiliations:** 1State Key Laboratory of Alternate Electrical Power System with Renewable Energy Sources, North China Electric Power University, Beijing 102206, China; 2Hebei Provincial Key Laboratory of Power Transmission Equipment Security Defense, North China Electric Power University, Baoding 071003, China

**Keywords:** plasma fluorination, GFRP, nano-doping, surface flashover, surface charge, trap energy levels

## Abstract

With the extensive application of glass fiber reinforced polymer (GFRP) in the field of high voltage insulation, its operating environment is becoming more and more complex, and the surface insulation failure has gradually become a pivotal problem affecting the safety of equipment. In this paper, nano-SiO_2_ was fluorinated by Dielectric barrier discharges (DBD) plasma and doped with GFRP to enhance the insulation performance. Through Fourier Transform Ioncyclotron Resonance (FTIR) and X-ray Photoelectron Spectroscopy (XPS) characterization of nano fillers before and after modification, it was found that plasma fluorination can graft a large number of fluorinated groups on the surface of SiO_2_. The introduction of fluorinated SiO_2_ (FSiO_2_) can significantly enhance the interfacial bonding strength of the fiber, matrix and filler in GFRP. The DC surface flashover voltage of modified GFRP was further tested. The results show that both SiO_2_ and FSiO_2_ can improve the flashover voltage of GFRP. When the concentration of FSiO_2_ is 3%, the flashover voltage increases most significantly to 14.71 kV, which is 38.77% higher than that of unmodified GFRP. The charge dissipation test results show that the addition of FSiO_2_ can inhibit the surface charge migration. By the calculation of Density functional theory (DFT) and charge trap, it is found that grafting fluorine-containing groups on SiO_2_ can increase its band gap and enhance its electron binding ability. Furthermore, a large number of deep trap levels are introduced into the nanointerface inside GFRP to enhance the inhibition of secondary electron collapse, thus increasing the flashover voltage.

## 1. Introduction

GFRP is a high-performance composite material with high toughness fiber as the skeleton, polymer as the filling matrix. Because of its excellent insulation, mechanical properties, heat resistance and corrosion resistance, it has been widely used in the field of high-voltage transmission. GFRP is not only the key material of composite insulator, composite cross arm, dry reactor and other equipment, but also shows broad application prospects in fully insulated tower and superconducting cable terminals [1,2]. However, with the continuous improvement of the voltage level of the power grid and the increasingly complex operation environment of the substation equipment, GFRP is facing more and more severe operating conditions, and its insulation performance is also facing new challenges [3,4].

GFRP will deteriorate and crack under the action of strong electric field during long—term charged operation, and the influence of external environment will cause the material to be affected by moisture, oxidation, pyrolysis and other problems. Finally, the defects such as fiber interface debonding and polymer matrix aging are caused [5,6]. These defects will not only reduce the mechanical properties of GFRP, but also cause equipment failures such as flashover and even insulation breakdown. A large number of charge traps will be formed on the surface and inside of the insulation material after degradation, and many charges will continuously sink and detrap under the action of external electric field [7,8]. Especially on the surface of materials, there is a serious distortion effect in electric field when the local charge accumulates to a certain extent. When the distortion electric field exceeds the critical value, it will induce flashover, seriously threatening the safe and stable operation of high-voltage transmission equipment [9]. Studies have shown that adjusting the charge trap of insulating materials by nano-doping and surface treatment can effectively improve the surface flashover voltage of materials [10,11].

With the continuous innovation of nano modification technology, the doping of functional nano fillers has become an important means of optimization and upgrading of composite materials [12,13]. The unique structures or textures formed by nano-particles in the composite significantly affect the macroscopic properties of the material. At the same time, the nano-interface structure and interface effect also profoundly affect the material properties [14]. In the study of modification of GFRP materials, functionalized nanoparticles can be used as a potential filler combined with fiber network, so as to improve the performance of GFRP. Burda et al. [15] modified GFRP by filling nano-SiO_2_@rubber core-shell structure, which significantly improved the toughness and electrical insulation properties of the composite. Meisak et al. [16] designed the Ni@C structure and doped it to modify GFRP to obtain a good electromagnetic shielding material. Baltopoulos et al. [17] dispersed carbon nanotubes in GFRP and constructed a packing network structure with good electrical conductivity, which realized nondestructive detection of internal defects in GFRP. Ghasemi Parizi et al. [18] proposed to use finite element method to study the nanointerface inside modified GFRP to analyze the interlayer shear behavior. Good nanostructure design is an efficient means to improve the comprehensive properties of GFRP. However, there is less research on improving the insulation properties of GFRP by nano-modification. It is still an urgent challenge to explore the influence of nano-interface on the insulation properties of GFRP and construct composite structures with excellent insulation properties.

In recent years, plasma technology has been gradually applied to the research on the modification of insulating materials. By bombarding the reactive substance into particles of high energy state, the grafting method is realized on the surface of the material, which has the advantages of high efficiency, low energy consumption and environmental protection. Yan et al. [19] modified BNNS by atmospheric bipolar nanosecond pulse DBD, and realized hydroxyl grafting with Ar as carrier gas and H_2_O as reactive gas, which improved the AC breakdown strength and thermal conductivity of BNNS/EP composites. Xie et al. [20] treated the surface of glass fiber with DBD, improved the interface bonding strength between glass fiber and epoxy resin, and enhanced the surface insulation performance of GFRP. Wang et al. [21] used low temperature atmospheric pressure plasma jet to ionize ethyl orthosilicate and deposit SiO_x_ on the surface of GFRP, which improved the shear strength between GFRP layers. This series of studies prove that plasma modification technology has a significant advantage in improving the performance of epoxy composites. At the same time, the use of plasma to achieve the functional modification of nano-filler, then to regulate the nanointerface structure inside the composite material has also aroused great interest of researchers. Plasma fluorination is an important and effective method to modify composite insulating materials. By using the strong electronegativity of fluorine, the trap property inside the material can be regulated, and the surface of the material can be given a very low secondary electron emission coefficient, which significantly improves the surface insulation strength of the composite. Ruan et al. [22] modified Al_2_O_3_ by DBD fluoridation and doped it with epoxy resin, increasing the surface flashover voltage of epoxy composite material by 34.7%. Shao et al. [23,24] grafted fluorine onto the surface of inorganic filler by DBD plasma technology, which improved the binding degree of filler and epoxy resin. At the same time, high-energy charge traps are introduced into the composite, which enhances the control ability of free charge and improves the insulation performance. Up to now, the research on plasma modification of the ternary system of fiber, polymer and nanoparticle is still in the initial stage, and the influence mechanism of the fluorinated nanointerface needs to be further explored.

Considering the advantages of nano-modification and plasma technology, this paper proposes to modify nano-SiO_2_ by DBD plasma fluorination and doping it into GFRP. The surface insulation properties of modified GFRP is further investigated, and the influence of nano-particles on the internal interface of GFRP is analyzed by microscopic characterization. In addition, the mechanism of flashover voltage increase is revealed by measuring the surface charge dissipation rate, surface conductivity, trap distribution and DFT calculation of nanoparticles.

## 2. Experiment

### 2.1. Materials

Bisphenol A epoxy resin (DGEBA, E-51), methyltetrahydrophthalic anhydride (MTHPA) and 2,4,6-tris (dimethylaminomethyl) phenol (DMP-30) are produced by Shanghai Resin Factory, Shanghai, China. E-class alkalifree glass fber provided by Langfang Anlang Sealing Materials Co., Ltd., Langfang, China. Nano SiO_2_, 50 nm, purchased from Shanghai Macklin Biochemical Technology Co., Ltd., Shanghai, China. 1H, 1H, 2H, 2H perfluorodecyltrimethoxysilane (fluorinated organic matter) (FAS-17), 97%, produced by Shanghai Macklin Biochemical Technology Co., Ltd., Shanghai China. CF_4_ and Ar, 99.99% purity, provided by North Special Gas Co., Ltd., Baoding, China.

### 2.2. Fluorination of SiO_2_ and Preparation of GFRP

Filler fluorination: DBD plasma method was used to fluorinate nano-SiO_2_ under atmospheric pressure, the platform as shown in Figure 1. The plasma fluorination process used in this paper is obtained from our previous research, which can graft nanofillers with high efficiency in a relatively short time [20,22,25]. Firstly, nano-SiO_2_ was pretreated with fluorine-containing organic matter. Appropriate amount of SiO_2_ filler was placed in a beaker, and a solution containing 2% FAS-17, 3% deionized water and 95% ethanol was added and fully stirred for 30 min. The mass ratio of SiO_2_ to FAS-17 was 1:0.5. Then it was placed in an incubator at 40 °C for full reaction for 24 h. The pretreated nano- SiO_2_ was placed in a quartz reactor and the reactor was placed between two round metal electrodes. Then, mixed gas of CF_4_ and Ar were fed into the reactor, and the flow rate of Ar was controlled by the flowmeter to be 2.5 slm and the flow rate of CF_4_ to be 0.4 slm. The plasma generator was turned on after the flow was stabilized. The voltage was adjusted to 7 kV and the center frequency was 50 kHz. It could be observed that orange filamentary spark appears in the reactor, and the fluorination treatment time was controlled within 15 min. After plasma fluorination, the filler was calcined in a tubular furnace at high temperature to make the residual FAS-17 fully volatilized. The calcination temperature was 280 °C and the treatment time was 4 h. The calcined powder was fully ground to obtain fluorinated SiO_2_ (FSiO_2_), and the fluorination effect was characterized by FTIR and XPS.

Preparation of composites: First, appropriate amounts of DGEBA and MTHPA were placed in a beaker and heated in a 60 °C oil bath for 30 min to blend well. Then nano- SiO_2_ before and after fluoride modification was added, and the doping concentration gradient of the filler was controlled to be 1%, 2%, 3%, 4% and 5%. Continue stirring for 30 min after adding the filler to make it evenly dispersed. Then the accelerator DMP-30 was added. The mass ratio of each component is DGEBA:MTHPA:DMP-30 = 100:80:1. The fully mixed material was soaked in glass fiber cloth layer by layer [26,27], and then hot pressed for 20 min at 140 °C and 10 MPa. After the GFRP material was formed, it was demoulded and placed in a drying oven, and then solidified for 10 h at 120 °C. Finally, GFRP samples with different formula systems were obtained. The GFRP modified by nano-SiO_2_ before and after fluorination were labeled as SiO_2_-GFRP and FSiO_2_-GFRP respectively, and their cross sections were characterized by SEM.

### 2.3. DC Surface Flashover Test

The GFRP samples were tested for negative DC flashover along the surface under atmospheric conditions using the uniform pressure rise method [26]. The experiment was conducted using a needle-needle electrode. The alignment of the tip of the needle electrode was adjusted and the spacing was controlled to be 7 mm. The center of the test surface of the sample was close to the lower surface of the electrode. Start the high voltage DC power supply and uniformly boost the voltage at a rate of 100 V/s until flashover discharge occurs. When flashover occurred, sharp discharge sound could be heard and blue arc could be observed on the surface of the sample. The voltage waveform of flashover could be collected by oscilloscope. At this time, the voltage was stopped immediately and the critical voltage value of flashover was recorded.

### 2.4. Surface Charge Dissipation Rate Test

In order to analyze the charge dispersion behavior and trap distribution characteristics on the surface of GFRP materials before and after modification, isothermal surface potential attenuation (ISPD) method was adopted in this paper to measure the surface charge of samples [27]. To ensure that the surface potential is zero before the test, the sample is ultrasonic cleaned and dried with deionized water. Then the sample was placed 5 mm below the corona needle, and the charging voltage was adjusted to −7 kV for 60 s. Then, the charged sample was placed 2 mm below the active capacitor probe for real-time potential data acquisition with a sampling frequency of 0.01 kHz and sampling time of 1600 s. Finally, the exponential decay function curve of surface potential with time could be obtained by plotting the obtained data.

### 2.5. Surface Resistivity Test

In the study of surface flashover of epoxy composite insulation materials, the surface resistivity of the material is also a crutial parameter that affects its surface insulation strength. In this paper, the three-electrode method was adopted to measure the surface resistivity of the sample [28]. During the measurement, the voltage was set at 500 V, and the pressure was applied until the data became stable. The surface current value *I_s_* was read through the Gillespie 6517 B electrometer, and the surface resistivity of the material could be calculated according to the following formula.
(1)$$R_{s}=\frac{2\pi U}{I_{s}\cdot d\cdot \ln\frac{D_{2}}{D_{1}}}$$
where *d* is the distance between the inner electrode and the protection electrode, *D*_1_ is the diameter of the inner electrode, *D*_2_ is the inner diameter of the protection electrode, and *U* is the test voltage. Each sample was tested 10 times. The sample was depolarized for 30 s before each measurement, and then the average value was calculated as the surface resistivity of the sample

## 3. Results

### 3.1. Characterization

The FTIR spectroscopy and XPS test results of SiO_2_ before and after fluoridation are shown in Figure 2. It can be seen from the Figure 2a that the anti-symmetric stretching vibration peak of Si-O-Si groups appears near 1115 cm^−1^ for SiO_2_ before and after fluorination. In addition, symmetrical stretching vibration peaks of Si-O-Si groups appear near 800 cm^−1^ and 475 cm^−1^ [29,30]. While the FSiO_2_ filler has an asymmetric stretching peak at 1152.7 cm^−1^ that can be attributed to -CF_2_ [31]. The electronegativity of the fluorine element is very strong, so the valence electrons will be bound near the nucleus of the fluorine atom, forming a charge center. The existence of this group is due to a large number of -CF_2_ active groups in the ionized product of CF_4_ and FAS-17, which are grafted on the surface of the filler by adsorption or direct bonding. In addition, there is a stretching vibration peak that can be attributed to C-F bond near 1216 cm^−1^ [32]. C-F is a strong polar covalent bond, and its structure is very stable. When a large number of -CF_x_ groups are grafted on the surface of the nano filler, the strong binding effect of fluorine on electrons will reduce the surface polarizability of the filler, which is conducive to improving the insulation performance of the material. The above characterization results showed that the -CF_x_ group was successfully grafted on the filler surface during the plasma fluorination process. It can be seen from the full spectrum of XPS that FSiO_2_ has a strong F 1 s peak at 689 eV compared with unfluorinated SiO_2_. In order to explore the existing form of fluorinated groups on the surface of SiO_2_, we have carried out peak-splitting treatment for element C, as shown in Figure 2d. The C 1 s peak is mainly decomposed into -CF_3_, -CF_2_, C-F and C-O peaks, which appear around 293.2 eV, 291.0 eV, 287.9 eV and 285.4 eV respectively, and the intensity relationship is -CF_2_ > C-O > -CF_3_ > C-F. It can be seen that CF_4_ and part of FAS-17 as fluorine sources are ionized step by step in the plasma discharge process, mainly to CF_2_^2+^ and CF_3_^+^, and combined with the active groups on the surface of nano SiO_2_, in the form of -CF_x_ grafted on the surface of filler.

The content ratio of main elements of SiO_2_ before and after fluoridation is shown in Table 1. The proportion of F element on the surface of FSiO_2_ reaches 40.66%, while the proportion of Si element and O element decreases about 20% compared with SiO_2_. It can be seen that fluorine-containing groups can be grafted onto the surface of nanoparticles well after being infiltrated by FAS-17 and treated by plasma fluorination. This makes the original polar groups such as hydroxyl group covered on the surface of the filler, or bombarded in the process of plasma discharge, resulting in fracture. The introduction of fluorine-containing groups can reduce the surface energy of the nano-filler and inhibit the agglomeration effect of the filler in the process of composite synthesis. At the same time, fluorine-containing groups introduced by plasma fluorination can bond with epoxy resin matrix to form fluorine-containing nanointerface, which can better regulate the degree of interfacial binding between epoxy resin, glass fiber and nano-SiO_2_, and introduce charge traps with higher energy levels at the interface.

The cross sections of GFRP materials of different systems were characterized, and the effects of SiO_2_ and FSiO_2_ on fiber interface were analyzed. It can be seen from Figure 3. that there are obvious pores in GFRP without doping filler, which is the specific manifestation of the poor binding degree between glass fiber and epoxy resin. Due to the lack of groups on the surface of glass fiber that can form stable chemical bonds with epoxy resin, it is difficult to avoid the existence of pores at the fiber interface inside conventional GFRP. The existence of these pores often increases the distortion degree of electric field, which is not conducive to maintaining the high insulation performance of the composites. The addition of nano-SiO_2_ can fill the pores between fiber and resin matrix to a certain extent, and the existence of active groups such as hydroxyl and carboxyl groups of silica can also enhance the degree of binding between epoxy matrix and glass fiber. However, as can be seen from Figure 3e, nano- SiO_2_ filler will have agglomeration effect, which also becomes the reason restricting the further optimization of the microstructure of GFRP materials. Therefore, there are still pores in the modified GFRP. As can be seen from Figure 3c, the addition of FSiO_2_ can make the binding between the epoxy resin and glass fiber compactness. There are no obvious pores in FSiO_2_-GFRP, which reduces the interface defects between epoxy matrix and glass fiber. According to the characterization results in Figure 3f, after plasma fluorination, FSiO_2_ shows good dispersion in GFRP, its agglomeration effect is significantly weakened, and the particle size of the aggregates is also much smaller than that of the unmodified system. The analysis shows that the etching roughness of FSiO_2_ surface treated by plasma fluorination will be significantly improved, which makes it form a mechanical action of occluding with fiber and matrix, and enhances the interface bonding strength. At the same time, the fluorine-containing groups on the surface of FSiO_2_ can also act as adhesives, which effectively improves the bonding degree between nano filler, glass fiber and epoxy resin.

### 3.2. Surface Flashover Voltage

The surface flashover voltage of modified GFRP with different formulations was tested 50 times in atmospheric environment. After eliminating the bad points through the boxplot, the trend diagram of flashover voltage changing with different packing concentrations was made, as shown in Figure 4. It can be seen from the figure that the DC flashover voltage along the surface of the unmodified GFRP material is 10.6 kV, and the flashover voltage of SiO_2_-GFRP and FSiO_2_-GFRP is higher than that of unmodified GFRP, and shows a trend of “first rising and then decreasing” with the increase of filler concentration. When the filler concentration of SiO_2_-GFRP was 2%, the flashover voltage was 12.59 kV, 18.77% higher than that of GFRP, and then the flashover voltage began to decrease. For FSiO_2_-GFRP, when the packing concentration is 3%, the maximum flashover voltage is 14.71 kV, which is 38.77% higher than that of GFRP, and the subsequent flashover voltage improvement effect is not obvious.

By comparing SiO_2_-GFRP and FSiO_2_-GFRP, it can be found that the flashover voltage improvement effect of the fluorinated material is more obvious. At the same time, it can be seen that the concentration of the optimal modification value of FSiO_2_-GFRP lags behind that of SiO_2_-GFRP. We believe that the introduction of fluorine-containing groups regulates the internal interface of GFRP, which further improves the insulation strength of the materials. When the filling concentration of unmodified nano-SiO_2_ increases gradually, the agglomeration will be serious, which will lead to the modification limited or even reduced. While the fluorination of SiO_2_ improves the dispersion of nanoparticles, so the range of its action concentration is also widened.

### 3.3. Surface Charge Dissipation Rate

Figure 5 shows the surface potential attenuation curves of SiO_2_-GFRP and FSiO_2_-GFRP with different concentrations. As can be seen from the figure, doping nano-SiO_2_ into GFRP can accelerate the surface charge dissipation rate of the composite. When the packing concentration is 2%, the dissipation rate increases the highest, and the final potential decreases about 36% compared with the initial potential. When the concentration of SiO_2_ increases further, the surface charge dissipation rate of GFRP becomes stable. After the fluorination of SiO_2_, the charge dissipation rate of FSiO_2_-GFRP is lower than that of GFRP. The charge dissipation ratio is about 2~7%, the higher the packing concentration, the slower the charge dissipation. This is because a large number of deep traps are formed inside modified GFRP by the introduction of fluorine containing groups. When charges migrate on the surface of the material under the action of external electric field, charges will be captured by the trap of high energy level and it is difficult to escape, thus suppressing the carrier migration rate on the surface of the material.

## 4. Results

### 4.1. Calculation of the State Density of SiO_2_

The trap energy distribution of composites is closely related to the properties of filler and nanointerface. Ding and Du et al. [33,34] proposed that the trap formation mechanism of materials can be analyzed by analogy of solid energy band theory [34]. In this paper, a spherical SiO_2_ model with a diameter of 7 Å was intercepted from the structure of the amorphous SiO_2_ model by using Materials Studio software, which contains 147 atoms. This is enough to reflect the nature of the material itself and meet the requirements of calculation. Combined with the characterization results of FTIR and XPS, part of the H atoms on its surface were replaced with -CF_2_ and -CF_3_ fluorinated groups to simulate the FSiO_2_, as shown in Figure 6. Based on DFT [35] and the generalized gradient approximation (GGA) [36,37], the PBE exchange-correlation potential is used to calculate the density of states (DOS) and the bandgap width of the model [38]. Frontier molecular orbitals are divided into HOMO and LUMO. The gap between the LUMO energy and the HOMO energy reflects the electron transition capacity. The frontier orbital level energy gap of the model is as follows:(2)$$E_{gap}=E_{LUMO}-E_{HOMO}$$

During the calculation, the quality is set to fine, and the maximum number of iterative steps and cycles is 1000. In order to speed up the convergence, the smudge method is used, with a value of 0.005.

Figure 7 shows the DOS of SiO_2_ and FSiO_2_, it can be seen that the energy of SiO_2_ and FSiO_2_ is mainly distributed in the range of −20~10 eV and reaches the maximum peak value at −18 eV and −5 eV, respectively. The bandgap width of SiO_2_ is about 4.986 eV, while that of FSiO_2_ is about 5.250 eV. The gap width, also known as the band gap, can reflect the transition ability of electrons. In the study of insulating materials, *E_gap_* is considered as a form of trap. The results show that the fluorinated grafting of SiO_2_ can increase the band gap width and make valence band electrons more difficult to transition into conduction band electrons, which is equivalent to introducing high level deep charge trap into the material and improving the insulation performance of FSiO_2_-GFRP.

### 4.2. Calculation of Trap Distribution of GFRP

Based on ISPD method, the surface trap distribution characteristics of GFRP with different systems were calculated [39]. By processing the surface potential attenuation data collected above, it can be found that there is a nonlinear discrete relationship between surface potential *U* and dissipation time *t*. Matlab software was used for curve fitting of surface potential and dissipation time, and the expression was:(3)$$U(t)=ae^{xt}+be^{yt}$$
where, *a*, *b*, *x*, and *y* represent the density peaks of different trap energy levels. By solving the fitted curve through Equations (3) and (4), the relationship between trap energy level *D* and trap density *I* can be obtained:(4)$$D=k_{B}T\ln(v_{ATE}t)$$
(5)$$I=t\frac{dU}{dt}$$
where, *T* is the ambient temperature during the test, Unit K; *k_B_* is Boltzmann constant; *ν_ATE_* is the electron escape rate, and the calculation formula is as follows:(6)$$\nu_{\mathrm{ATE}}=\frac{{{(k}_{B}T)}^{3}}{dh^{3}v^{2}}$$

According to the measured surface potential of SiO_2_-GFRP and FSiO_2_-GFRP with different concentrations, the corresponding surface trap distribution is calculated, as shown in Figure 8. Combined with charge dissipation, it can be seen that SiO_2_ and FSiO_2_ affect the charge dissipation behavior of the composites mainly by changing the trap energy level of GFRP. For SiO_2_-GFRP, when the packing concentration is higher than 2%, the shallow trap density increases obviously, and the deep trap energy level also decreases to a certain extent, which is the main reason for the acceleration of the surface charge dissipation rate. When the filling concentration is 1%, the introduction of a small amount of nano-filler will affect the crosslinking structure of the original epoxy resin. This will lead to the destruction of the original charge transmission channel, so the acceleration effect of nano SiO_2_ on charge is difficult to reflect. When the packing concentration increases, a large number of nanointerfaces formed inside GFRP bring more shallow trap levels, so the carrier migration ability is enhanced. The trap energy level of FSiO_2_-GFRP shows an obvious upward trend with the increase of filler concentration. This indicates that the ability of trapping and binding charge on the surface of FSiO_2_-GFRP material is enhanced, which is basically consistent with the dissipation of charge. The analysis shows that the grafted fluorine-containing groups on the surface of FSiO_2_ form a large number of fluorine-containing interface regions in GFRP. And the strong electron binding ability of fluorine makes it easy to capture charges, while the captured charges are difficult to be excited again. At the same time, the charge center formed by the captured carriers also has strong binding ability, which is reflected in the enhancement of trap energy level on the surface of FSiO_2_-GFRP.

Combined with the calculation results of the state density of SiO_2_, it is found that grafting fluorine on the surface of SiO_2_ can increase its band gap and enhance its electron binding ability, which is consistent with the above analysis. It is also observed that when the filling concentration of FSiO_2_ increases to 5%, the trap energy level decreases. This is because the filling limit of nano-filler in epoxy resin is basically about 5%, and excessive doping leads to agglomeration and stacking of filler, thus forming more physical defects, namely shallow traps. Therefore, the binding ability of the composite to the surface charge is slightly weakened, and the ability to inhibit the occurrence of flashover along the surface is weakened, and the flashover voltage begins to drop.

### 4.3. Surface Resistivity

In order to further analyze the influence of nano-SiO_2_ before and after fluorination on GFRP flashover voltage, the surface resistivity of GFRP was tested, and the results are shown in Figure 9. As can be seen from the figure, the surface resistivity of SiO_2_-GFRP shows an inverted “N” type change trend with the increase of filler concentration, while the surface resistivity of FSiO_2_-GFRP shows a trend of “first increasing then decreasing”. When the concentration of SiO_2_ is 1%, the resistivity of GFRP is relatively high, about 19 × 10^15^ Ω·cm, which is consistent with the analysis conclusion of the above trap. With the further increase of packing concentration, the surface resistivity of GFRP decreases significantly, which is related to the introduction of active groups and interface regions. With the further increase of filler concentration, the insulating property of nano-SiO_2_ itself began to play a role, so the resistivity rose to a certain extent. However, when the packing concentration continues to increase, the agglomeration and packing stacking effect enhance the carrier transport ability, so that the surface resistivity of GFRP increases slightly. When using FSiO_2_ as filler, the surface resistivity of FSiO_2_-GFRP is higher than that of SiO_2_-GFRP. When the concentration of FSiO_2_ increases to 3%, the resistivity of FSiO_2_-GFRP increases to about 23 × 10^15^ Ω·cm, which is mainly dependent on the introduction of a large number of fluorine-containing nanointerfaces in the material. This effect is highly consistent with the above analysis. With the further increase of filler concentration, F-SiO_2_ also appeared agglomeration phenomenon, resulting in a slight decrease in surface resistivity, the surface resistivity is about 9 × 10^15^ Ω·cm when the concentration is 5%.

According to literature [40], the charge generated by the negative polarity high-voltage electrode dissipates through three ways: internal attenuation, surface migration, and neutralization with hetero-sign ions. When the surface charge migrates along the surface of the composite, it will be affected by the trap in the surface, and the trapping and detrapping of the charge will directly affect the migration rate of the charge. Furthermore, the surface flashover voltage of composite insulating materials is affected. The influence of GFRP structure and trap on the surface charge dissipation process is shown in Figure 10.

For SiO_2_-GFRP, nano-filler and fiber form a network conducive to charge dissipation inside the composite. At the same time, the deep traps introduced by the nano-interface are low in energy level, and more shallow traps are introduced. The trapped charges are easier to detrap, and it is not easy to restrain the occurrence of surface discharge by binding the charges. In addition, the surface resistivity of the composite is low, which is conducive to the surface charge migration, avoiding a large amount of charge accumulation on the surface in a short time, and increasing flashover voltage to a certain extent. The flashover voltage increase of FSiO_2_-GFRP is more significant. The analysis shows that after plasma fluorination, FSiO_2_ promotes the bonding strength between nano particles, fibers and resin matrix, which optimizes the microstructure of the material and improves the comprehensive property. On the other hand, the grafted fluorine-containing groups on the surface of FSiO_2_ increase the band gap and deepen the trap energy level on the surface. The energy required for charge detrapping increases, which inhibits the secondary electron emission and electron avalanche process in the development of flashover. Therefore, the DC flashover voltage along the surface of the material is significantly increased.

## 5. Conclusions

In this paper, FAS-17 was used to pretreat nano-SiO_2_, and the filler was modified by DBD plasma fluorination. The influence of nano- SiO_2_ before and after fluorination on the surface insulation performance of GFRP was further studied. The results show that fluorine is mainly grafted on the surface of nano-SiO_2_ in the form of -CF_x_. After plasma fluorination, FSiO_2_ can enhance the binding degree of epoxy resin matrix and glass fiber, which is conducive to improve the comprehensive performance of GFRP materials. The DC flashover voltage of GFRP can be improved by nano-SiO_2_ before and after modification. Among them, the flashover voltage of GFRP modified by FSiO_2_ doping increases more significantly. When the filling concentration of FSiO_2_ is 3%, the flashover voltage of GFRP reaches 14.71 kV, which is 38.77% higher than that of unmodified GFRP. The surface charge dissipation rate of GFRP samples was tested. It was found that SiO_2_ can accelerate the charge dissipation rate of GFRP surface, while FSiO_2_ can inhibit the charge dissipation of the material surface. By calculating the state density of nano-SiO_2_ and the trap distribution of composites before and after modification, it is found that grafting fluorine on SiO_2_ surface can increase its band gap and enhance its electron binding ability. This is equivalent to increasing the energy level of deep trap inside GFRP, improving the ability to inhibit secondary electron emission during the development of flashover discharge, and thus increasing the surface flashover voltage of GFRP. The research in this paper provides a new idea for improving the surface insulation performance of GFRP in the field of high voltage insulation, expands the application prospect of high performance GFRP, and promotes the application of plasma technology in the modification of composite insulation materials.

## Figures and Tables

**Figure 1 nanomaterials-13-00906-f001:**
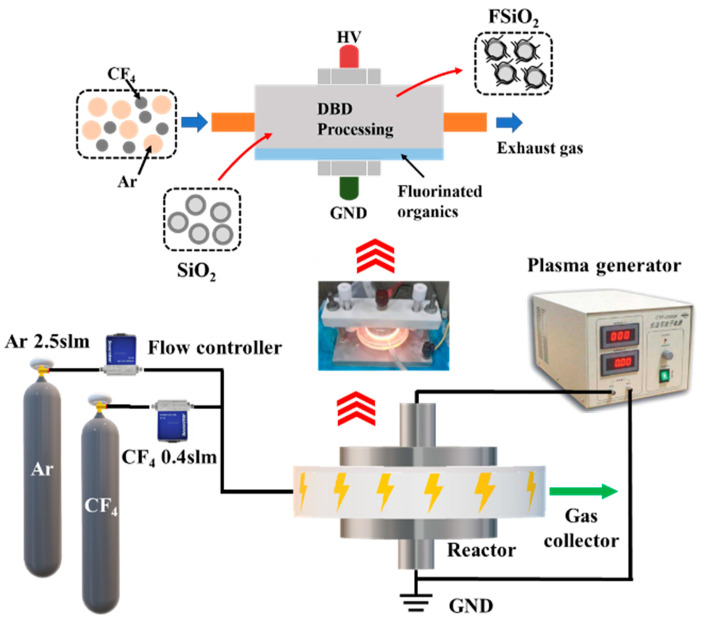
Plasma fluorination platform.

**Figure 2 nanomaterials-13-00906-f002:**
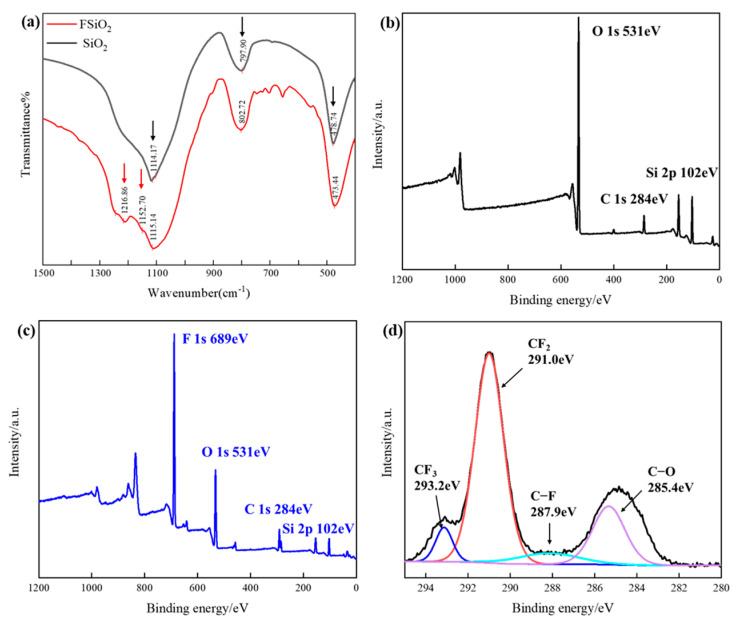
FTIR and XPS characterization of SiO_2_ before and after fluoridation. (**a**) FTIR. (**b**) XPS of SiO_2_. (**c**) XPS of FSiO_2_. (**d**) The peak-splitting result of element C.

**Figure 3 nanomaterials-13-00906-f003:**
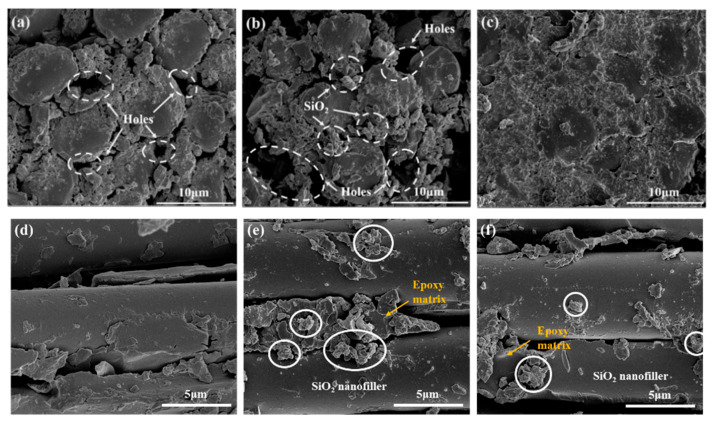
SEM characterization of axial direction of GFRP section. (**a**) GFRP. (**b**) SiO_2_-GFRP. (**c**) FSiO_2_-GFRP. SEM characterization of radial direction of GFRP section. (**d**) GFRP. (**e**) SiO_2_-GFRP. (**f**) FSiO_2_-GFRP.

**Figure 4 nanomaterials-13-00906-f004:**
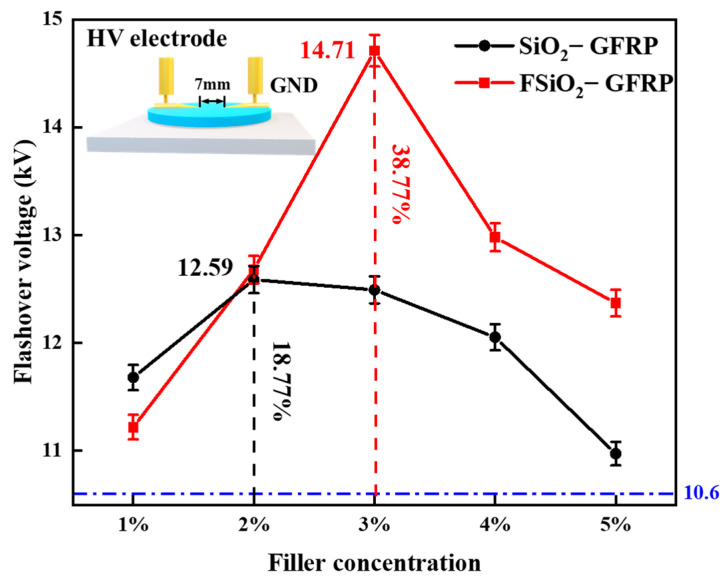
Flashover voltage curve with packing concentration.

**Figure 5 nanomaterials-13-00906-f005:**
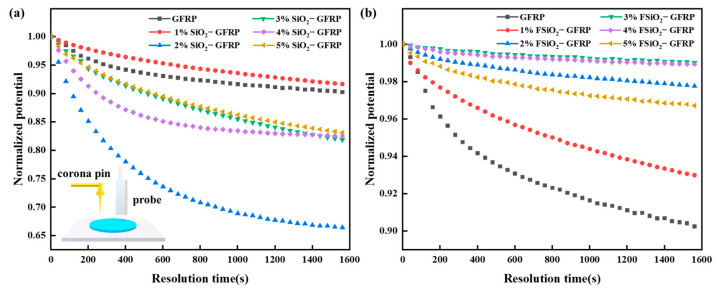
Potential decay curve of GFRP samples. (**a**) SiO_2_-GFRP. (**b**) FSiO_2_-GFRP.

**Figure 6 nanomaterials-13-00906-f006:**
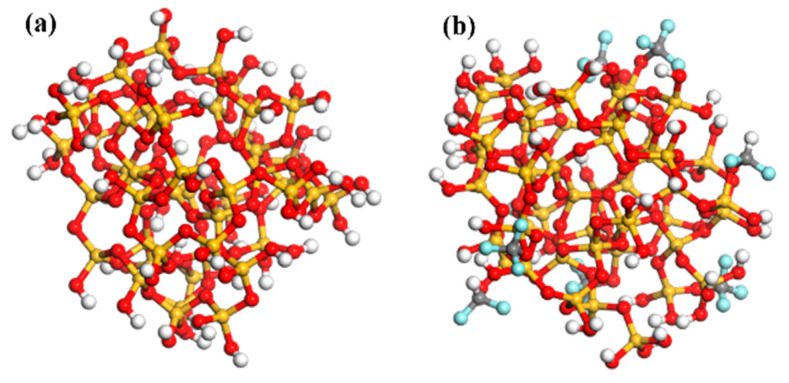
Molecular model. (**a**) SiO_2_. (**b**) FSiO_2_.

**Figure 7 nanomaterials-13-00906-f007:**
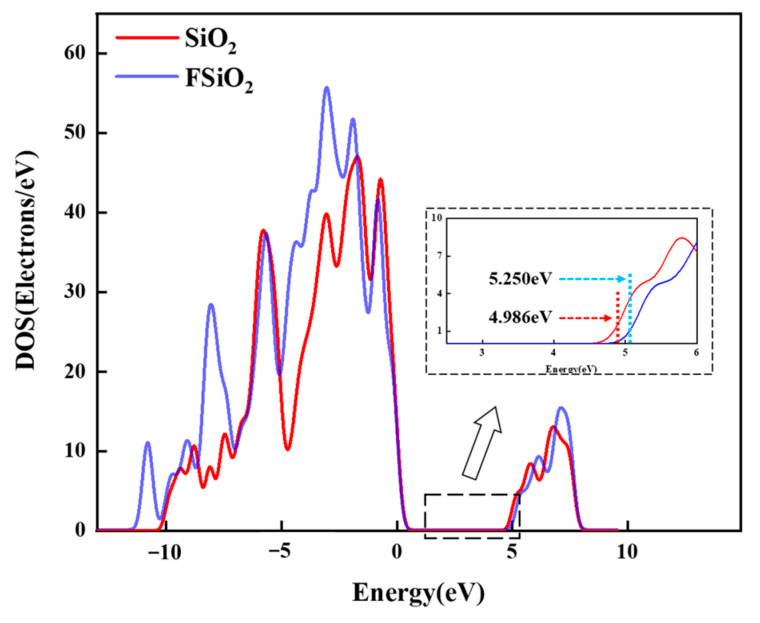
DOS diagram of SiO_2_ before and after fluorination.

**Figure 8 nanomaterials-13-00906-f008:**
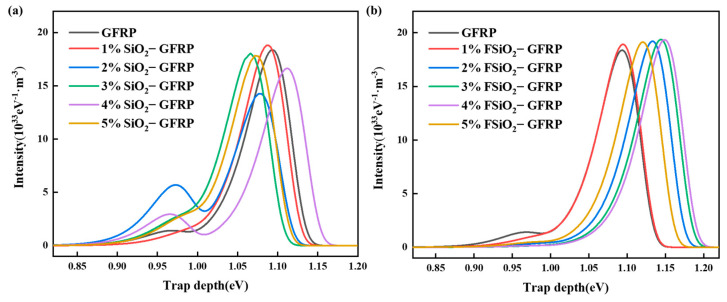
Trap energy level and density distribution of GFRP. (**a**) SiO_2_-GFRP. (**b**) FSiO_2_-GFRP.

**Figure 9 nanomaterials-13-00906-f009:**
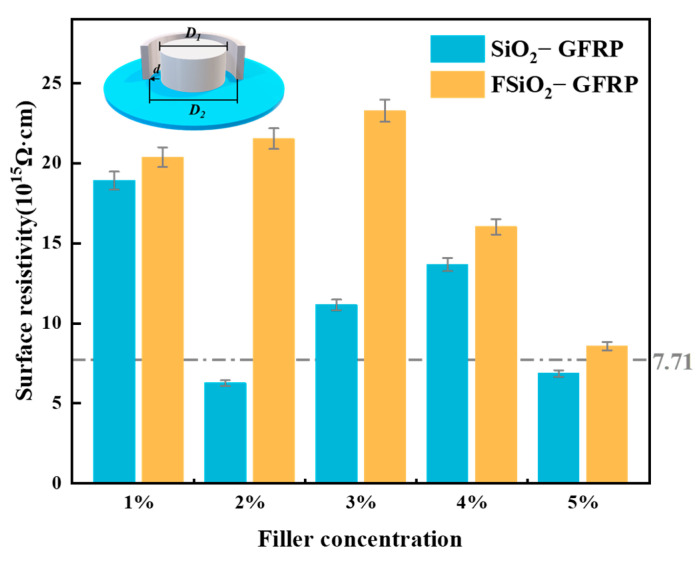
Surface resistivity of GFRP samples.

**Figure 10 nanomaterials-13-00906-f010:**
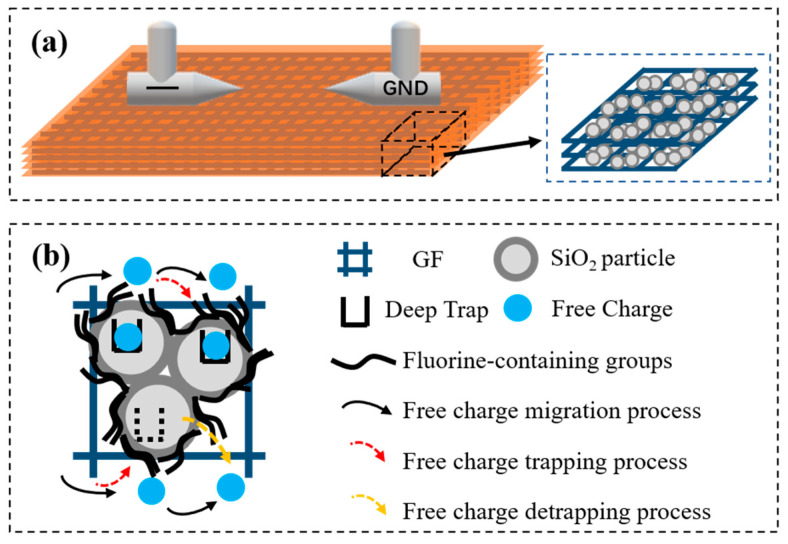
Schematic diagram of charge transport mode. (**a**) schematic diagram of modified GFRP structure. (**b**) influence mechanism of trap on charge transport.

**Table 1 nanomaterials-13-00906-t001:** Proportion of elements on SiO_2_ surface before and after fluorination.

Type of Filler	Si (%)	O (%)	F (%)
SiO_2_	32.17	55.60	0
FSiO_2_	13.61	23.36	40.66

## Data Availability

Data will be provided upon request, please contact us if necessary. The email is as follows: duan_ncepu@163.com.

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
