# Peer review of "Plasma Fluorinated Nano-SiO2 Enhances the Surface Insulation Performance of Glass Fiber Reinforced Polymer"

_nanomaterials, 2023, doi:10.3390/nano13050906_

Round 1

Reviewer 1 Report

Dear Authors,

it could be interesting to specify the nature of some chemical bonding mentioned in FTIR paragraph  (ex: CF2 asymetric stretching : 1152 cm-1)

Could you give more dteails about the deconvolution performed in Fig. 2d?

Could you specify the nature of chemical bond suggested with expoy resin?

Have you performed EDX analysys coupled with your SEM measurements ? Acartography of nanoparticle dispersion would be interesting.

You mention that FSi02 improves the bonding degree and that nanoparticles act as an adhesive? Have you performed mechanical tests to support this conclusion?

To justify your hypotheses according different concentration of nanoparticles you suppose an agglomeration. Have you taken SEM images for each composite to verify this?

On Fig. 4, I supposed that the flashover of GFRP is 10.6 kV. It is not clear. Please add this information in the main text.

Concerning the model of the part 4.1 you wrote "SiO2 model with a diameter of 5A......., part of the H atoms on its surface were replaced with-CF2 and -CF3"

==> How have you chosen the diameter?

==> the number of H atoms replaced have been determined according the values of Fig. 2d?

Author Response

Thank you for your review. We have uploaded the reply and modified content as a word file. Please check it.

Reviewer 2 Report

The authors address plasma processing of nano-SiO2 to enhance surface insulation performance in glass fiber reinforced polymer. The authors gave detailed analysis with scientific point of view on the fluorination effect for the nano-SiO2 in the performance of GFRP. I consider the topic original or relevant in the field with detailed studies addressing the further possible development in the field. Application of fluorine plasma of SiO2 can be significant for application in glass fiber reinforced polymers.

Authors can further analyze the change in fluorinated SiO2 particles properties with change in fluorination duration, plasma power, selection of fluorine gas flow rates, chamber pressure.  

I think the tables and figures are clear. The figure 1 can be bigger in size to have a clear idea of the plasma process and experimental process.

The authors reported on the plasma fluorination of nano-SiO2 enhancing the surface insulation performance of GFRP.

I would like to suggest to include full form of 'GFRP' as Glass Fiber Reinforced Polymer in the title of the manuscript. 

Further, please check English of the manuscript. 

I recommend minor revision of the manuscript for consideration of publication.

Author Response

(The authors gave the same response as above.)
